# Development of a Rapid and Sensitive CANARY Biosensor Assay for the Detection of Shiga Toxin 2 from *Escherichia coli*

**DOI:** 10.3390/toxins16030148

**Published:** 2024-03-14

**Authors:** Christina C. Tam, Yangyang Wang, Wen-Xian Du, Andrew R. Flannery, Xiaohua He

**Affiliations:** 1Foodborne Toxin Detection and Prevention Research Unit, Western Regional Research Center, Agricultural Research Services, United States Department of Agriculture, 800 Buchanan Street, Albany, CA 94710, USA; christina.tam@usda.gov (C.C.T.); wen-xian.du@usda.gov (W.-X.D.); 2Smiths Detection, 2202 Lakeside Boulevard, Edgewood, MD 21040, USA; yangyang.wang@smiths-detection.com (Y.W.); andrew.flannery@smiths-detection.com (A.R.F.)

**Keywords:** Shiga toxins, STEC, biosensor, CANARY^®^, detection, B-cell based assay, immunoassay, food safety

## Abstract

Shiga-toxin-producing *Escherichia coli* (STEC) causes a wide spectrum of diseases including hemorrhagic colitis and hemolytic uremic syndrome (HUS). The current Food Safety Inspection Service (FSIS) testing methods for STEC use the Food and Drug Administration (FDA) Bacteriological Analytical Manual (BAM) protocol, which includes enrichment, cell plating, and genomic sequencing and takes time to complete, thus delaying diagnosis and treatment. We wanted to develop a rapid, sensitive, and potentially portable assay that can identify STEC by detecting Shiga toxin (Stx) using the CANARY (Cellular Analysis and Notification of Antigen Risks and Yields) B-cell based biosensor technology. Five potential biosensor cell lines were evaluated for their ability to detect Stx2. The results using the best biosensor cell line (T5) indicated that this biosensor was stable after reconstitution with assay buffer covered in foil at 4 °C for up to 10 days with an estimated limit of detection (LOD) of ≈0.1–0.2 ng/mL for days up to day 5 and ≈0.4 ng/mL on day 10. The assay detected a broad range of Stx2 subtypes, including Stx2a, Stx2b, Stx2c, Stx2d, and Stx2g but did not cross-react with closely related Stx1, abrin, or ricin. Additionally, this assay was able to detect Stx2 in culture supernatants of STEC grown in media with mitomycin C at 8 and 24 h post-inoculation. These results indicate that the STEC CANARY biosensor developed in this study is sensitive, reproducible, specific, rapid (≈3 min), and may be applicable for surveillance of the environment and food to protect public health.

## 1. Introduction

Shiga toxin (Stx) was originally described as an infectious toxin produced by *Shigella dysenteriae* serotype 1. It is now clear that a number of serotypes of *Escherichia coli* (*E. coli*) also produce one or more Shiga-like toxins. Microbiologists now often use Shiga toxin and Shiga-like toxin interchangeably. From 2009 to 2021, there were 1019 reported Shiga-toxin-producing *Escherichia coli* (STEC) outbreaks resulting in 14,010 illnesses, 2218 hospitalizations, and 43 deaths in the US alone [1]. STEC has many different virulence effectors; however, Shiga-like toxins (Stxs) are the predominant agents responsible for causing hemorrhagic colitis and hemolytic uremia syndrome in severe cases [2,3,4]. Stxs from *Escherichia coli* are ribosomal inactivating proteins that can be divided into two different groups: Stx1 (subtypes: Stx1a, Stx1c, Stx1d, and Stx1e) and Stx2 (Stx2a–i, and Stx2k) [5,6,7,8,9,10]. Expression of Stx1a, Stx2a, Stx2c, or Stx2d by STEC strains is correlated more with human disease, but illnesses have been shown to be caused by the other subtypes [11,12,13,14,15]. Stxs can be stable under extreme conditions, even in those that would normally kill the organism; thus, having biologically active Stx in food products can cause disease [16,17,18,19,20,21]. Early, rapid, and sensitive detection of STEC/Stx will greatly reduce morbidity and mortality since there are no FDA-approved therapeutics to treat the complications that can arise from STEC infections.

There are numerous methods used to detect Shiga toxins and/or the presence of Stx genes including PCR, ELISA, LC-MS, mouse bioassays, cell-free assays, cell-based assays, biosensors, etc. [22,23,24,25,26,27,28,29,30,31,32,33,34,35,36,37,38,39,40]. Each of these detection assays have their own advantages and disadvantages. The in vivo mouse bioassay is the “gold standard” to detect active toxins but can be laborious, expensive, time-consuming, and requires specialized facilities and trained personnel. A mouse bioassay developed previously was able to detect Shiga toxins with a limit of detection (LOD 290 ng/kg, 290 ng of Stx2 per kilogram of mouse body weight) and models all aspects of Stx intoxication including antibody protection [41]. ELISAs have been developed to detect Stxs with high sensitivity (LOD ≈ 0.025 ng/mL) in a shorter time span (2.5 h compared to days for the mouse bioassay). In vitro cell-free translation assays have been developed to detect biological active Stx with high sensitivity and reproducibility but cannot model all aspects of Stx intoxication [26]. Recently, a cell-based cytotoxicity assay was developed that detected Stx even when the STEC titer was less than 0.4 CFU/mL in water but the read-out was two days, thus impairing early diagnosis and treatment [23].

CANARY^®^ (Cellular Analysis and Notification of Antigen Risks and Yields) is a cell-based biosensor technology. Immortal B-cell lines express antigen-specific antibodies on the surface of cells and contain aequorin from *Aequoria victoria*. This technology has been shown to detect the pathogens *Yersinia pestis*, Vaccinia virus, Venezuelan equine encephalitis virus, *E. coli* O157:H7, and *Bacillus anthracis* [42]. It has also been tested against a variety of detection platforms (mostly lateral flow) using the potential bioterror threats *Bacillus anthracis* and ricin with LODs of 10^3^ spores/mL and 3 ng/mL, respectively [43]. The authors found that compared to all the other commercially available kits, the CANARY^®^ Zephyr platform was 4 orders of magnitude more sensitive for detecting *B. anthracis* and was the most sensitive for ricin. Additionally, this platform was used to detect botulinum neurotoxin serotype A in a variety of complex food matrices with LODs in ng/mL using very small volumes and times (in minutes) [44].

Due to the complexity involved in STEC outbreaks, multiple technologies may be required for early diagnosis and treatment of STEC infection. In this study, we sought to develop a rapid and sensitive CANARY^®^ B-cell-based biosensor assay and assess its feasibility in the detection of Stx in PBS buffer and STEC cultures.

## 2. Results

### 2.1. Development and Characterization of a CANARY^®^ B-Cell Based Assay to Detect Shiga Toxin 2

Figure 1 shows the STEC CANARY^®^ biosensor assay. Biosensors expressing membrane-bound antibodies that are specific to epitopes of Stx2 were assessed for their ability to bind to and detect Stx2 toxoid and toxin. The binding of the antigen of interest directly by the antibodies on the biosensors’ surface leads to an intracellular calcium influx that activates a signal transduction event causing the emission of light. The luminometer detects the light output, which is expressed as relative light units (RLU) over time (120 s, read every second). When a low concentration of Stx2 is present and binds to some of the antibody-displaying B-cell surface, the signal transduction event initiated will be low, and thus the light emission will be low (Figure 1A). However, in the event of a high concentration of Stx2 present as shown in Figure 1B, multiple surface antibodies bind to Stx2, thus initiating a higher signal transduction event that amplifies the signal leading to high light emission. Thus, this B-cell based biosensor assay is dose-dependent and its output reflects this.

Five biosensor cells lines (T1–T5) were generated by cloning the heavy and light chains of Stx2a monoclonal antibodies (mAbs) developed in house previously [45] into B-cell lines. Stx2a toxoid was spiked into an assay buffer provided by the manufacturer. The detection capability of each STEC CANARY biosensor cell line was evaluated. Figure 2A depicts the performance of each CANARY cell line in the detection of Stx2a toxoid. In total, 200 μL of each experimental sample was added to microcentrifuge tubes and then 20 μL of reconstituted room temperature STEC biosensors in foil were added to the cap of the tube. The reaction was initiated by mixing the sample with the biosensor by centrifugation for 5 s inside the instrument, put in the luminometer, and read. The results were shown graphically in real-time and recorded for each second for 120 s. T1 (black circle), T2 (red circle), and T5 (blue circle) cell lines produced a rapid and high signal when detecting the assay buffer spiked with 2 µg/mL Stx2a toxoid as compared to T3 (green circle) and T4 (purple circle) cell lines, which were both flat. The T5 biosensor cell line had the highest RLU with the fastest onset as compared to T1 or T2, so we decided to focus on optimizing the T5 cell line for Stx2 detection. As depicted in Figure 2B, the sensitivity of the T5 biosensors to Stx2 was determined. Toxoid was serially diluted in assay buffer ten-fold from 400 ng/mL to 0.004 ng/mL. A dose-dependent response of the STEC T5 biosensors to a decrease in the toxoid concentration was seen. At high concentrations (40–400 ng/mL), rapid onset of the RLU signal was seen (a sharp rise, peak, and decrease in signal). As the Stx2 concentration decreases (from 40 ng/mL downwards), a shift to the right and flattening of the curve can be observed. At 0.04 ng/mL, a slight peak can be seen compared to 0 ng/mL, the assay buffer control, from 80 s onwards. However, no differences can be determined with the assay buffer control at 0.004 ng/mL.

### 2.2. Precision of the STEC Biosensor Assay: Repeatability, Reproducibility, and Shelf-Life

The STEC T5 biosensors were evaluated for their repeatability, reproducibility, and shelf-life, as shown in Figure 3. A set of biosensors were reconstituted on day 1 and an aliquot was used to test for the intra-day variability of the assay. The rest of the biosensors were stored at 4 °C wrapped in foil for use on day 5 and day 10. Toxoid was serially diluted ten-fold as before in the assay buffer but two additional doses of 0.2 ng/mL and 0.1 ng/mL dose were added to the assay since the RLU curve flattened significantly between 0.4 ng/mL and 0.04 ng/mL in Figure 2B. As shown in one representative graph in Figure 3A (left), similar RLU curves were determined. We observed a sharp rise, peak, and flattening of the curve at 400 ng/mL and shifting of the RLU curves to the right and gradual flattening as the toxoid concentration decreased, similar to those depicted in Figure 2B. In Figure 3A (right), we have expanded and focused on the curves with the doses between 0.4 ng/mL to 0.04 ng/mL with 0 ng/mL (assay buffer control). The 0 ng/mL assay buffer control curves are generally pretty flat from 20 s to 120 s with the lines hovering near 200 RLU. At 0.4 ng/mL, there is a gradual rise in the RLU from 50 s onwards with a peak at ≈1600 RLU and then flattening. With both 0.2 ng/mL and 0.1 ng/mL, both show a gradual but slower increase as compared to 0.4 ng/mL with peaks of ≈800–900 RLU and ≈500–600 RLU, respectively. For 0.04 ng/mL, there was evidence of variability between the duplicates but the trend for this dose was relatively flat and slightly above the assay buffer control. With the assay buffer control curves having a background baseline of ≈200 RLU, we can estimate a tentative limit of detection of ≈600 RLU, which is 3 × 0 ng/mL (assay buffer control). With this estimated cut-off ≈ 600 RLU, we can estimate that the LOD of this biosensor assay is ≈0.1–0.2 ng/mL. Additionally, a proprietary algorithm is being developed to more precisely determine the limits of detection.

The reproducibility (inter-day variability) of the biosensor assay was investigated on day 5 and day 10. We show that the curves and trends for the assay were similar on day 5 (Figure 3B, left) and day 10 (Figure 3B, right). The peak RLUs were similar on day 1 and day 5. On day 10, similar but slightly decreased peak RLUs were seen in the assay for doses from 400 ng/mL to 4 ng/mL. However, the doses from 0.4 ng/mL to 0.004 ng/mL showed curves that were more reduced compared to days 1 and 5. On day 10, only the 0.4 ng/mL curve was above 600 RLU and thus determined to be the LOD. These experiments have determined that the STEC T5 biosensors were repeatable, reproducible, and stable up to 10 days after reconstitution with the caveat regarding the slightly decreased LOD on day 10 compared to days 1 and 5.

### 2.3. STEC Biosensor Specifically Recognizes Shiga Toxins

Since we have established that the STEC T5 biosensor assay was highly sensitive, we wanted to investigate the specificity of the assay with Stx2 subtypes, Stx1, and other ribosome-inactivating proteins like abrin and ricin. All Stx2 subtypes, Stx1a, abrin, and ricin were diluted to 100 ng/mL in assay buffer before the addition of biosensors and initiation of the Zephyr software version 3.0. This high dose was used due to the consistent and reproducible high signals seen in the ng/mL levels when tested (Figure 2 and Figure 3) and to eliminate any uncertainty associated with the use of lower toxin concentrations. As graphically displayed in Figure 4, STEC T5 biosensors detect most Stx2 subtypes with high RLU signals. Table 1 depicts the results of the toxoid/toxin panel specificity further. The STEC T5 biosensors were able to detect Stx2a, Stx2b, Stx2c, Stx2d, and Stx2g, but not Stx2e and Stx2f. This is maybe due to the relatively larger variations in amino acid sequences of Stx2e and Stx2f vs Stx2a because the antibody used in the T5 biosensor was made with the Stx2a antigen. Importantly, this assay did not show cross-reactivity with Stx1a, abrin, or ricin.

### 2.4. Detection of Native Stx2 Produced by STEC in Media

We tested the STEC biosensors’ ability to detect natively expressed Stx2 from culture supernatants since this would be the scenario that would occur when testing for foodborne outbreaks. The initial studies showed that four biosensor cell lines were able to detect Stx2 from overnight cultures of *E. coli* 0121:H19 grown in buffered peptone water (BPW) + novobiocin and the signal was enhanced with the addition of mitomycin C as has been reported. Figure 5 shows the results for the initial testing of T5 biosensors using overnight culture samples. Overnight bacterial cultures were split into the following groups for the biosensor assay: (1) overnight cultures (200 μL); (2) cell pellets from 200 μL overnight cultures reconstituted in 200 μL assay buffer; (3) 50 μL of supernatants from centrifuged overnight cultures + 150 μL assay buffer; and (4) 200 μL of supernatants from centrifuged overnight cultures. It was found that the addition of mitomycin C enhanced Stx expression and resulted in higher RLU signals (filled circles) regardless of the sample group tested compared BPW + novobiocin alone (open circles). The most important result from Figure 5 was that 200 μL of supernatants from centrifuged overnight cultures (red filled circle) gave the highest RLU signal in comparison to the other mitomycin C treated samples like the overnight cultures (filled light blue circle), cell pellet (filled black circle), and 50 μL supernatants from centrifuged overnight cultures + 150 μL assay buffer (filled yellow circle).

**Therefore**, the overnight culture supernatants were selected for the following biosensor assays. We next evaluated the potential use of this assay to detect Stx2 from STEC culture supernatants as early as 8 h (for early detection and product recall) post-inoculation. A variety of O157 and non-O157 STEC strains along with appropriate control strains were grown in Luria broth (LB) with mitomycin C. As depicted in Table 2 and Figure 6 for the 8 h samples, Stx2 was not detected in the negative control strains ATCC25922 (O6, *stx-*) and RM7103 (O45, *stx-*). The rest of the strains that express Stx2 or Stx2 Stx1 (RM12788) showed variable RLU signals that were significantly higher than the negative control strains with the highest signal corresponding with the 8 h supernatants from RM5856.

Interestingly, the biosensors detected lower light emission signals in the 24 h culture supernatant samples from all the *stx+* STEC strains (Figure 7A) compared to the 8 h supernatant samples (Figure 6 and Table 2). These results suggested that there may be some matrix effect from the 24 h culture supernatant samples that either inhibited biosensor binding with the toxin and/or the signal transduction and signal amplification that occurs after antigen–antibody binding. To overcome this negative effect on the biosensor assay with the 24 h culture supernatants, the samples were diluted 1:10 into assay buffer and the assay was repeated. Unsurprisingly, the 1:10 dilution of the 24 h culture supernatants with assay buffer relieved the matrix effect (Figure 7B). All the *stx+* STEC strains had relatively higher RLU curves than the negative control strains ATCC25922 and RM7103, which were flat.

## 3. Discussion

The detection of Shiga toxins employs numerous approaches including antibody-based assays, mass spectrometry, cell-based assays, cell-free assays and in vivo mouse assays. All the current technologies have advantages and disadvantages. No single technology is adequate for applications in all settings, including clinical, food safety, or environmental settings; thus, new technology platforms should be assessed for their ability to improve the current methods for the detection of Shiga toxins.

This CANARY^®^ system utilizes a B-cell based biosensor system to detect Stx2 without the need of immunomagnetic bead capture as seen with BoNT/A [44], which shortens the assay time down to 3 min from the addition of sample and biosensors to the read out of luminescence and final determination of positive or negative for the sample tested. This is better than most traditional methods. The CANARY^®^ system is also portable exclusive of cell culture growth, suitable for field studies because it consists of a laptop, a small centrifuge, and a small luminometer that can fit in a suitcase [43]. Yet, its portability does not sacrifice utility, as the assay also uses small volumes (200 μL) to facilitate multiple sample analysis for precious samples.

This is the first demonstration using the CANARY^®^ biosensor system to detect Shiga toxins. It was shown that the STEC T5 biosensor is a useful tool that can be applicable in food safety surveillance and environmental testing for STEC with excellent sensitivity (LOD ≈ 0.1–0.2 ng/mL), specificity, short assay time, reliability, minimal sample preparation, small volumes (≈200 μL), and shelf-stable up to day 10. This technology detects Stx2 much faster as compared to other detection methods such as ELISAs (hours), LC-MS (hours), cell-free translation assays (1.5 h), cell-based cytotoxicity assay (days), and the mouse bioassay (days). Additionally, the assay performed well with purified toxins present in assay buffer and natively expressed toxins in culture supernatants by, as early as 8 h post-inoculation, which would help with product recall, early diagnosis, and initiation of therapeutics for suspected STEC illnesses. Further optimization of the biosensor protocol may yield improvements in the assay’s sensitivity and applicability in complex food matrices as needed for outbreak scenarios. Additionally, this platform could be further developed into rapid detection kits for all subtypes of Stx1 and Stx2.

## 4. Conclusions

The STEC biosensor assay described in this study detects both Stx2 spiked in assay buffer and from STEC bacterial culture supernatants (even at 8 h post-inoculation) with high sensitivity and specificity. The rapid assay time (≈3 min) and small volumes needed (≈200 μL) makes this detection technology a potent qualitative tool in environmental and food safety surveillance programs.

## 5. Materials and Methods

### 5.1. Reagents

The Stx toxoids and holotoxin were produced and purified in-house (USDA) as described previously [22,45,46,47]. Abrin and ricin were purchased from Toxin Technology (Sarasota, FL, USA) and Vector Laboratory (Burlingame, CA, USA), respectively. STEC biosensors and assay buffer were produced by Smiths Detection. Stx toxins were stored at −20 °C, bacterial supernatants were stored at −80 °C, while abrin and ricin were stored at 4 °C before use. STEC biosensors were stored in liquid nitrogen and thawed immediately before use. The CANARY^®^ Zephyr detection system consists of a laptop, a small microcentrifuge (SCILOGEX D1008, Rocky Hill, CT, USA) and a luminometer (Sirius L Tube Luminometer TITERTEK-Berthold, Pforzheim, Germany).

### 5.2. Biosensor Engineering

Five Stx2a monoclonal antibodies were generated previously using a recombinant Stx2a E167Q toxoid expressed in BL-21(DE3) pLysS strain [45]. The sequences from the mAbs were determined. The DNA sequence coding the heavy and light chains from each of the five mAbs were synthesized, transformed, transfected, and screened with antibiotic selection into B-cell biosensors. The development of these biosensors was due to a Cooperative Research and Development Agreement between Smiths Detection and United States Department of Agriculture, Agricultural Research Service. The inherent details regarding biosensor development and biosensors themselves are proprietary information. Initial testing with each of the five different STEC biosensors with Stx2a toxoid and STEC cultures were performed.

### 5.3. Biosensor Reconstitution

Aliquots of B-cell biosensors were stored in liquid nitrogen for long-term storage. Assay buffer was removed from the refrigerator (4 °C) and allowed to reach room temperature prior to use (≈1 h). One vial of biosensors (or multiples as needed for experiments) were thawed for 2 min in a 37 °C water bath with gentle swirling. The biosensors were transferred to a sterile microcentrifuge tube and centrifuged at 200× *g* for 5 min at 4 °C to remove freezing media. The supernatant was aspirated out and the cell pellet was resuspended in 1 mL of assay buffer per vial of biosensor thawed with gentle pipetting. Cell viability was determined with trypan blue staining and cell counting. Viable (trypan blue excluded cells) biosensors were reconstituted to the defined working concentration in assay buffer, wrapped with foil, and either kept at room temperature for 30 min before use or kept in foil at 4 °C for up to 10 days for long-term use.

### 5.4. STEC Biosensor Assay for Stx2a Toxoid Detection

Stx2a toxoid was retrieved from −20 °C and thawed at room temperature in a biological safety cabinet. From the working Stx2a toxoid stock, an initial dilution to 400 ng/mL was made in assay buffer in a total volume of 500 μL and allowed to disperse for 20 min at room temperature. Subsequently, ten-fold serial dilutions were made to 0.004 ng/mL in assay buffer. 200 μL of each dilution was added to microcentrifuge tubes and then 20 μL of reconstituted room temperature STEC biosensors in foil were added to the cap of the tube. Zephyr initiation causes the reaction tube to be centrifuged for 5 s before placement into the luminometer to be read immediately. Luminescence was recorded every second for 120 s and displayed live in a graph. Each sample was read in duplicate. A proprietary algorithm to determine positive and negative responses based on the signal-to-noise and curve characteristics dependent on up to 28 coefficients is being developed. Samples with RLUs below 600 were considered as negative or not detected, which was defined by RLUs from assay buffer controls (about 200) multiplying 3.

### 5.5. Shelf-Life of STEC Biosensors

Shelf-life stability of STEC biosensors were performed similarly to Section 5.4 with the following modifications. Two vials of T5 biosensors from liquid nitrogen were thawed and resuspended in assay buffer at 5.0 × 10^5^ cells/mL in assay buffer and wrapped with foil. One biosensor aliquot was used for Day 1 while the rest were stored at 4 °C for use on Day 5 and Day 10. Tenfold serial dilutions were performed with Stx2a toxoid as in Section 5.3, but additional concentrations of 0.2 ng/mL and 0.1 ng/mL were evaluated.

### 5.6. STEC Biosensor Specificity Assay

Stock solutions of Stx2a, Stx2b, Stx2c, Stx2d, Stx2e, Stx2f, Stx2g, Stx1a, ricin and abrin were taken out from 20 °C and thawed at 4 °C in a biological safety cabinet. Five hundred μL of 100 ng/mL of each toxin solution were prepared in assay buffer and allowed to disperse for 20 min at room temperature. The assays were then performed as described in Section 5.3.

### 5.7. Initial E. coli 0121:H19 BPW + Novobiocin +/− Mitomycin C Study

*E. coli* 0121:H19 (NR-17630, *stx1- stx2*) from BEI was grown in Buffered Peptone Water (BPW) with novobiocin (BN) in the presence or absence of mitomycin C at 100 ng/mL. 200 μL in total volume of sample was added with 20 μL of biosensors. The CANARY system was initiated, and luminescence was recorded. The samples tested were the following from the overnight cultures: (1) overnight cultures; (2) cell pellet of overnight cultures resuspended in 200 μL assay buffer; (3) 50 μL supernatant after centrifugation + 150 μL assay buffer; and (4) 200 μL of overnight supernatants centrifuged.

### 5.8. Detection of Stx from STEC Supernatants with T5 Biosensor

Eight bacterial strains were streaked from glycerol stocks onto Luria Broth (LB) agar plates in a biosafety hood and incubated at 37 °C overnight. The strains of interest were as follows: *E. coli* O157:H7 strains: RM1913, RM10058; *E. coli* O103 strain: RM1046; *E. coli* O111 strain: RM12788 (*stx1 stx2*,); *E. coli* O121 strain: RM6848; *E. coli* O145 strain: RM9872; *E. coli* O6 strain: ATCC25922 (*stx-*); and *E. coli* O45 strain: RM7103 (*stx-).* Single colonies from each plate were inoculated into 15 mL Falcon tubes (Corning, Corning, NY, USA) containing 10 mL LB + 100 ng/mL of mitomycin C and grown at 37 °C with 200 rpm shaking for 24 h. At 8 H and 24 H post-inoculation, four mL of each bacterial culture were collected. After centrifugation at 10,000× *g* for 10 min at 4 °C, the supernatants were filtered through a 0.2 µm filter. The cleared filtered culture supernatants were stored at −80 °C before testing with the biosensor. The biosensor assay was run the same way as in Section 5.4, except that the 24 H supernatants required a dilution of 1:10 with assay buffer to abrogate the matrix inhibition effects. Duplicate samples per strain supernatants were tested in the assay. Two independent experiments were performed (ND: not detected; +: detected).

## Figures and Tables

**Figure 1 toxins-16-00148-f001:**
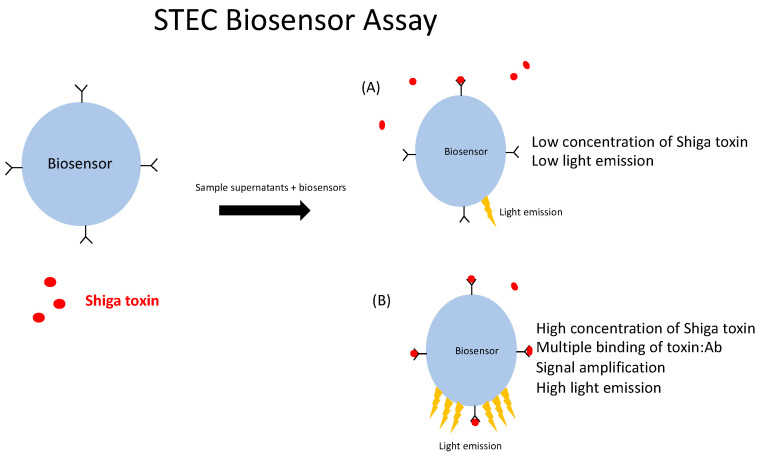
Schematic of STEC CANARY^®^ biosensor assay. B-cell biosensor cell lines expressing antibodies on their cell surface bind to the antigen of interest. (**A**) Binding of Shiga toxin 2 to the surface antibodies on the biosensors induces a signal transduction cascade triggering light emission. Low concentrations of Stx2 causes low light emission (low RLU). (**B**) As more of the antigen binds to the antibody receptors on the cell surface, amplification of the signal is induced, and more light is emitted as seen when there is a high amount of Stx2.

**Figure 2 toxins-16-00148-f002:**
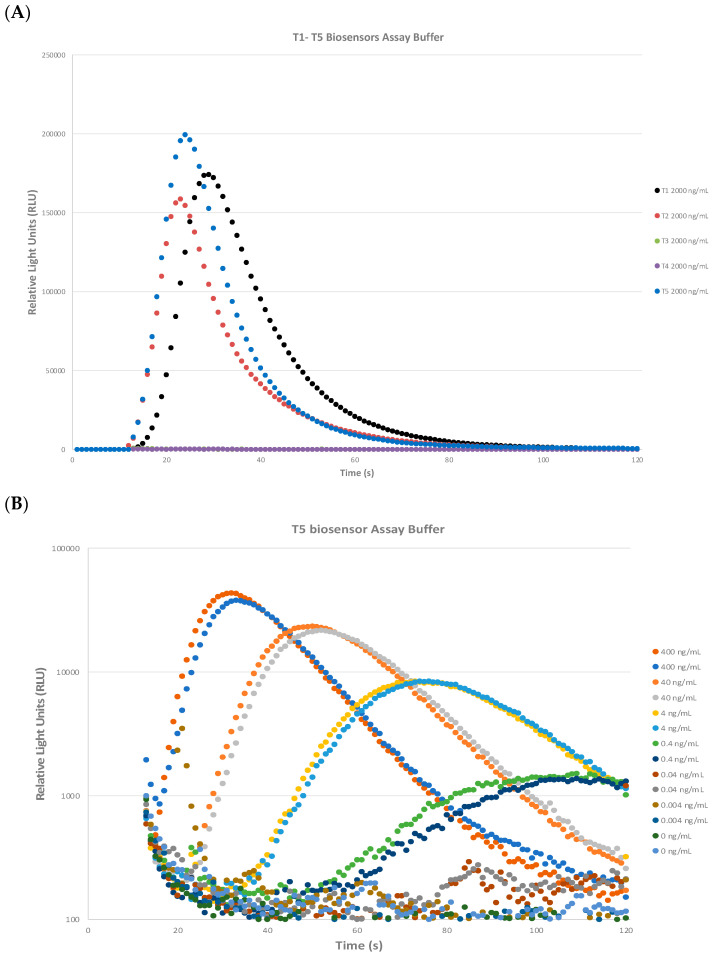
STEC CANARY^®^ biosensors detect Stx2 in assay buffer with high sensitivity in a concentration-dependent manner. (**A**) Five biosensor (T1–T5) cell lines were generated and evaluated for their ability to detect Stx2a toxoid in assay buffer. T1 (black circle), T2 (red circle), and T5 (blue circle) biosensors elicited a rapid and significant high signal at 2,000 ng/mL compared to T3 (green circle) and T4 (purple circle) cell lines. One representative data set is presented. Two independent experiments were performed. (**B**) STEC T5 biosensors detect Stx2a toxoid in assay buffer with high sensitivity in a concentration-dependent manner. Toxoid was serially diluted in assay buffer from 400 ng/mL to 0.004 ng/mL and each independent experiment (n = 2) was performed with duplicate samples. One representative result is shown. A dose-dependent shift to the right and flattening of the RLU curve as the Stx concentration is decreased.

**Figure 3 toxins-16-00148-f003:**
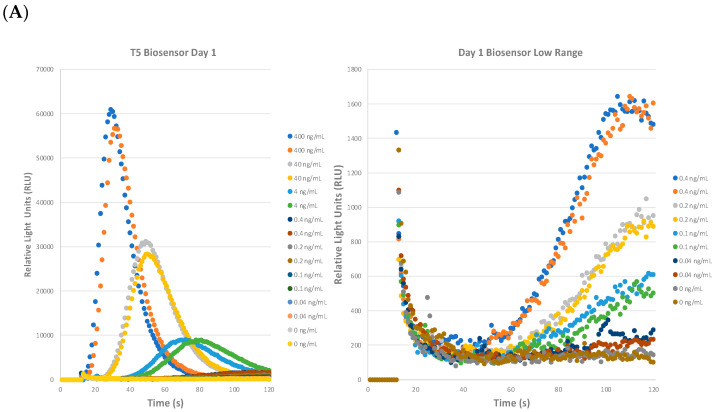
STEC biosensors have a stable shelf-life and detect Stx2 with high sensitivity. Biosensors were reconstituted on day 1 and used to detect Stx2a toxoid serially diluted in assay buffer on days 5 and 10. (**A**) The repeatability (intra-day variability) of the STEC biosensors was evaluated. Biosensors were reconstituted on day 1 and used to detect Stx2a toxoid serially diluted in assay buffer. (**B**) The reproducibility (inter-day variability) of the STEC biosensors was evaluated on days 5 and 10. Duplicates were performed for each independent experiment per dose. Two independent experiments were performed consecutively.

**Figure 4 toxins-16-00148-f004:**
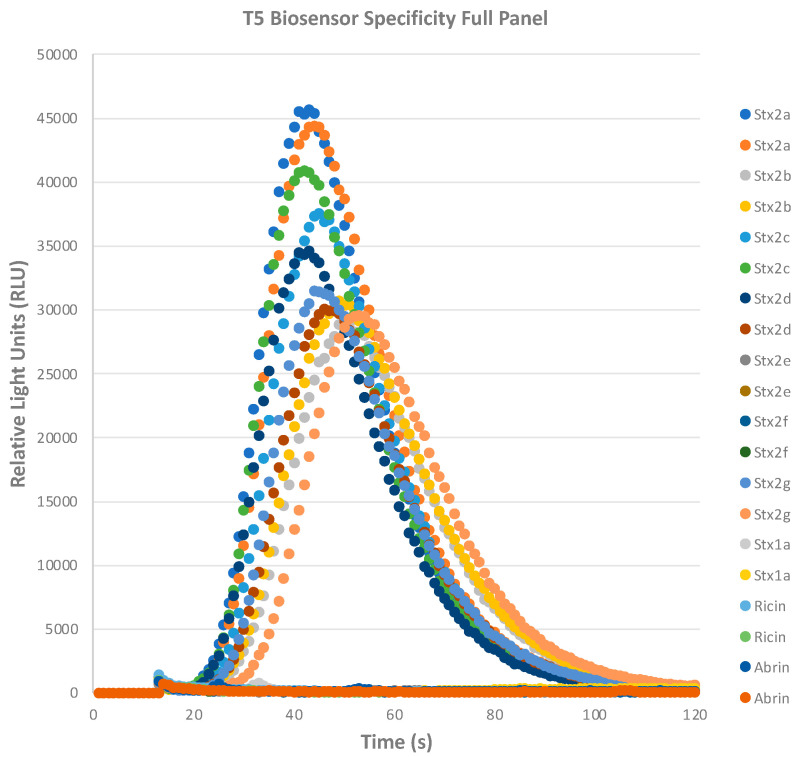
Specificity of STEC biosensors. Stx toxoids, abrin, and ricin were diluted to 100 ng/mL in assay buffer above the limit of detection of the assay. Duplicates were measured with biosensors for luminescent signal.

**Figure 5 toxins-16-00148-f005:**
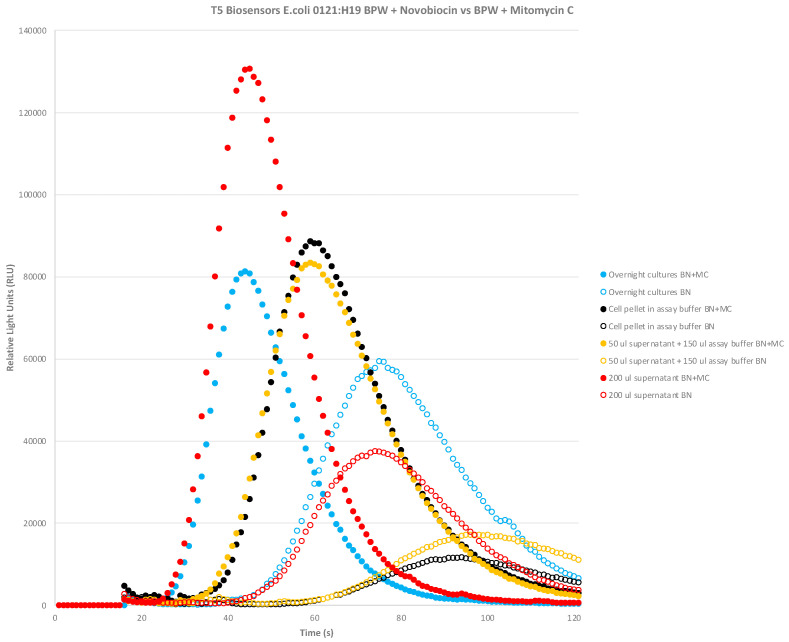
T5 STEC CANARY^®^ biosensors detect natively expressed Stx2 from *E. coli* O121:H19. *E. coli* O121:H19 (*stx1- stx2+*) was grown in buffered peptone water (BPW) with novobiocin (BN) in the presence (filled circles) or absence of mitomycin C (empty circles). One representative data set is presented. Negative controls (media, stx- strains) were not depicted on the graph due to their extremely low RLU signal in comparison to the samples.

**Figure 6 toxins-16-00148-f006:**
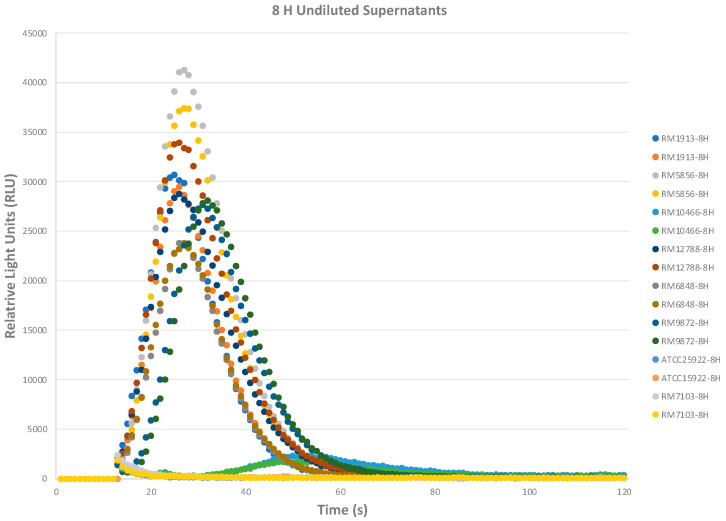
8 H STEC cultures express Stx2 that can be detected with the biosensor assay. The graph depicts the various RLU values from the biosensor assay with undiluted culture supernatants 8 H post-inoculation. Duplicate samples per strain supernatants were tested in the assay. Two independent experiments were performed.

**Figure 7 toxins-16-00148-f007:**
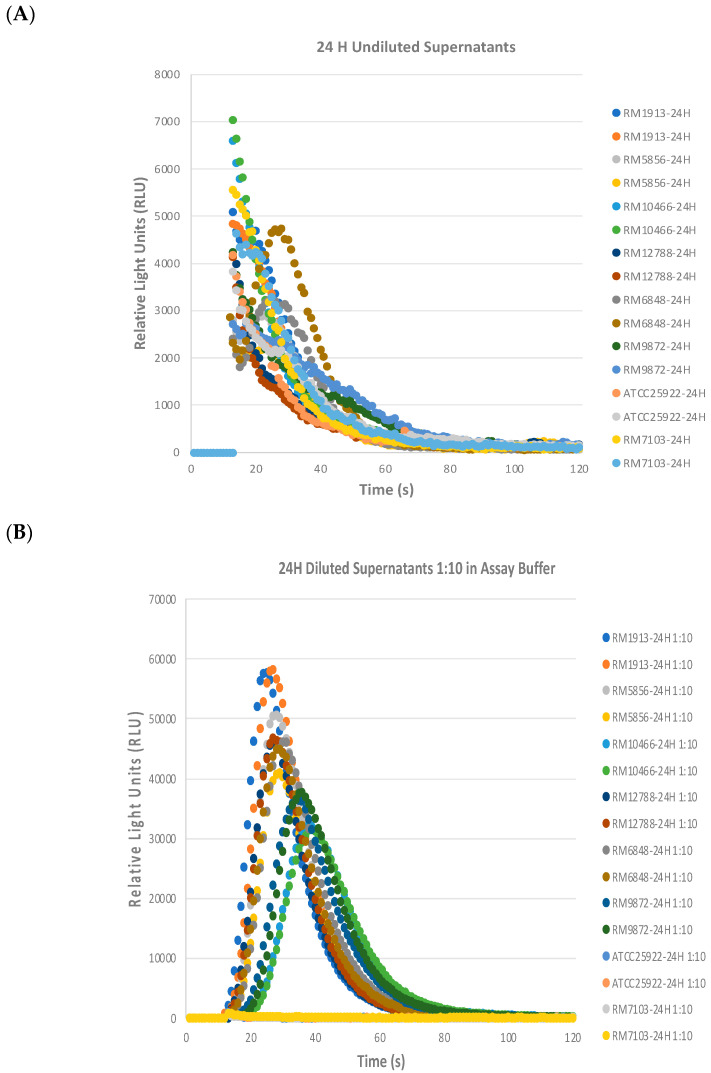
Matrix effects from 24 H STEC culture supernatants on the biosensor assay. (**A**) Results from un-diluted 24 H STEC culture supernatants. (**B**) Results from 1:10 diluted 24 H culture supernatants. Duplicate samples per strain supernatants were tested in the assay. Two independent experiments were performed.

**Table 1 toxins-16-00148-t001:** STEC T5 CANARY biosensors specifically recognize a subset of Shiga toxin 2 subtypes but not Stx1, abrin, or ricin.

Toxin	Detection
Stx2a	+
Stx2b	+
Stx2c	+
Stx2d	+
Stx2e	ND
Stx2f	ND
Stx2g	+
Stx1a	ND
Abrin	ND
Ricin	ND

ND, not detected (RLUs ≤ 600); +, detected (RLUs > 600).

**Table 2 toxins-16-00148-t002:** Detection of Stx2 in culture supernatants at 8 h post-inoculation using STEC CANARY biosensors.

Strains	Type	Detection
RM5856	O157:H7	+
RM1913	O157:H7	+
RM10466	O103	+
RM12788	O111 (*stx1* and *stx2*)	+
RM6848	O121	+
RM9872	O149	+
ATCC25922	O6 (*stx-*)	ND
RM7103	O45 (*stx-*)	ND

ND, not detected (RLUs ≤ 600); +, detected (RLUs > 600).

## Data Availability

The data presented in this study are available in this article.

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
