# Peer review of "Development of a Rapid and Sensitive CANARY Biosensor Assay for the Detection of Shiga Toxin 2 from Escherichia coli"

_toxins, 2024, doi:10.3390/toxins16030148_

Round 1

Reviewer 1 Report

Comments and Suggestions for Authors

The manuscript described an efficient method for detecting Stx2 using CANARY biosensor system. There are no major issues with the analysis and the study was well-designed and carefully conducted. I have a few comments (minor revisions) for the authors to address:

1. Is any threshold value for URL which was considered as ND? It would be interesting to include this information in the text or in tables 1 and 2.

2. Lines 235 – 237: “Interestingly, the biosensors detected lower light emission signals in 24H culture supernatant samples from all the stx+ STEC strains (Figure 6A) as compared to the 8H supernatant samples (Table 1).” First, would the table cited here be Table 2?

Second, it is difficult to conclude this comparison as there are no RLU values from the Stx2 detection test of culture supernatants at 8 hours. It would be interesting to insert the RLU values considered as + or ND, or present them in a graph.

3. Lines 366 – 368: “All strains detected signal except for the negative control strains ATCC25922 and RM7103. RM5856 had the highest signal out of all the stx2+ strains in comparison.” This sentence could be excluded because it is already very well described in the Results section.

Author Response

     Thank you for the enthusiastic support for the manuscript and all the helpful comments and suggestions for Toxins manuscript-2847557. 

Reviewer 1 comments:

  1. Is any threshold value for URL which was considered as ND? It would be interesting to include this information in the text or in tables 1 and 2.

--- Thank you for your comment and suggestion related to the ND threshold value. We have tentatively assigned samples whose RLUs throughout the 2 min read with curves that are below 600 RLU as negative/not detected. This assignment was based on the assay buffer controls being about 200 RLU and multiplying 3x this signal to give a tentative cutoff of 600 RLU.  The proprietary algorithm for this assay has not been developed, it would need to include many different characteristics (up to 28 factors) and not just the maximum RLU seen. We have defined RLUs below 600 as negative in the M & M section (lines 433-434) and in Table 1 and Table 2.

  1. Lines 235 – 237: “Interestingly, the biosensors detected lower light emission signals in 24H culture supernatant samples from all the stx+ STEC strains (Figure 6A) as compared to the 8H supernatant samples (Table 1).” First, would the table cited here be Table 2?

Second, it is difficult to conclude this comparison as there are no RLU values from the Stx2 detection test of culture supernatants at 8 hours. It would be interesting to insert the RLU values considered as + or ND, or present them in a graph.

---Thank you for commenting on the error regarding the reference to Table 1. We have revised the manuscript accordingly to reflect the correct reference table (Table 2).  Additionally, we have added a new Figure 6 that depicts the RLU results from the undiluted 8 H culture supernatant graphically. Accordingly, the old Figure 6 with the 24 H culture supernatants is now Figure 7.

  1. Lines 366 – 368: “All strains detected signal except for the negative control strains ATCC25922 and RM7103. RM5856 had the highest signal out of all the stx2+ strains in comparison.” This sentence could be excluded because it is already very well described in the Results section.

--- Thank you for noting this. We have removed these two sentences from the section.

Reviewer 2 Report

Comments and Suggestions for Authors

This manuscript describes the development of a cell-based biosensor for the analysis of Shiga toxin 2. This study used b-lymphocytes transformed cells that display anti-shiga antibodies on their surface. This technology, called CANARY, has been previously published for other applications, such as botulinum toxin and Salmonella analysis.

The manuscript is very well written and it fits the aims and scopes of the journal. However, in my opinion, the results are not adequately presented and additional experiments are needed.

To avoid confusion, I suggest using the same units throughout the manuscript and graphs, for example ng/mL.

To study the variation of the signal as a function of analyte concentration, the maximum signal obtained in each case must be depicted vs the analyte concentration. Triplicate experiments are necessary and error bars must be included. This is particularly important to determine the LOD. This parameter is usually calculated as 3 times the standard deviation at zero dose of analyte.

Figure 3 needs to be redone. The variation of RLU as a function of analyte concentration must be depicted. Moreover, in my opinion, a signal decrease is observed with time. Which is the value of LOD at day 10?

The study depicted in figure 5 needs a negative control without Stx2.

The authors claim that this is a portable, rapid assay; however, cell cultures of at least 8 h are required. How many cells were used in the initial inoculum? This value will determine the time required to detect the toxin in the culture. Can these cell cultures be incubated outside the lab for portability? These points need to be revised throughout the manuscript.

Only STEC culture supernatants were analyzed. Environmental or food samples must be included.

Author Response

Reviewer 2 comments:

  1. To avoid confusion, I suggest using the same units throughout the manuscript and graphs, for example ng/mL.

--- Thank you for the suggestion. We have modified the manuscript and graphs to ng/mL to be consistent in units.

  1. To study the variation of the signal as a function of analyte concentration, the maximum signal obtained in each case must be depicted vs the analyte concentration. Triplicate experiments are necessary and error bars must be included. This is particularly important to determine the LOD. This parameter is usually calculated as 3 times the standard deviation at zero dose of analyte.

--- Thank you for comment and suggestion. We agree that most detection assays should be displayed as suggested above with the maximum signal depicted vs the analyte concentration. However, the CANARY biosensor assays are a semi-quantitative assay whose response curve(s) over time with different analyte concentrations are of great importance. The response curve’s shape, peak, time to initiation of peak, and many other curve characteristics are factored in to develop a proprietary algorithm using machine learning to determine positive or negative signals. This is due to the inherent binding of antigen to the multiple receptors on the B-cell biosensors and the signal amplification that results from this binding. We have independently performed experiments will all relevant negative and positive controls in duplicate on multiple days with the biosensors to determine its repeatability and reproducibility. We have conservatively estimated the LOD as 3 times the zero dose of analyte (200 RLU x 3= 600 RLU) instead of the conventional 3 times standard deviation of zero analyte plus zero analyte because of the low analyte curves seen in this assay. We have added a sentence in the M & M section (lines 433-434) to define the LOD used in this study.

  1. Figure 3 needs to be redone. The variation of RLU as a function of analyte concentration must be depicted. Moreover, in my opinion, a signal decrease is observed with time. Which is the value of LOD at day 10?

--- The Figure 3 suggestion has been noted. The purpose of Figure 3 is to show the stability/shelf-life of the biosensors and the reproducibility of the assay over time. We believe that showing the Day 1 biosensor results in Figure 3A graphically with both the full panel and the inset for low concentrations to see in detail the curve characteristics as depicted with signal vs time per concentration is important to understand this assay and its differences with other conventional detection assays where maximal RLU signal is usually used to determine LOD. The day 5 and 10 graphs are important to see any shifts due to potential biosensor assay degradation which we see a little bit in day 10 where the estimated LOD as currently used (600 RLU and below are considered negative, 3x media/buffer control over the entire 2 min of the assay) shifts to 0.4 ng/ml instead of the 0.1 – 0.2 ng/mL that we can determine from day 1 and day 5. We have modified the text in abstract and relevant portions (lines 186-192) in the manuscript.

  1. The study depicted in figure 5 needs a negative control without Stx2.

--- We have performed all experiments with their relevant positive and negative controls including buffer/media control/control strains without Stx2. We have not shown these controls in Figure 5 because those curves will be suppressed all the way to the bottom as seen in the other figures and did not want to distract from the point of the figure to compare the results with or without mitomycin C treatment.

  1. The authors claim that this is a portable, rapid assay; however, cell cultures of at least 8 h are required. How many cells were used in the initial inoculum? This value will determine the time required to detect the toxin in the culture. Can these cell cultures be incubated outside the lab for portability? These points need to be revised throughout the manuscript.

--- Thank you for the comment. We agree that the cell culture inoculum and growth time to express Stx2 will affect the biosensor assay. The initial inoculum for all cell culture experiments in this study was one single colony from a freshly streaked plate containing the strains of interest. We wanted to clarify that the CANARY system exclusive of cell culture growth is potentially portable since the components of the assay are relatively small as we have added in the manuscript (see lines 356-357). We can envision food outbreak samples (which may include high amounts of Stxs) or potentially food samples from inspection can be used directly with the CANARY system without enrichment initially and potentially with/in conjunction with enrichment. Obviously, the amount of Stx in the samples and expression of Stx from different STEC strains can be variable. But identification of STEC strains at 8 hours post inoculation using CANARY biosensor is quite impressive because most assays require overnight culturing to identify positive strains.  Our subsequent manuscript on the applicability of this assay will explore this issue.

  1. Only STEC culture supernatants were analyzed. Environmental or food samples must be included.

 Thank you for the insightful suggestion regarding environmental or food samples to be evaluated in this assay.  Those experiments are not in the current scope of this manuscript which describes the initial development of the STEC assay.  We have another manuscript that relates to the applicability of this STEC biosensor assay for detection of Stx/STEC in contaminated food samples.

Reviewer 3 Report

Comments and Suggestions for Authors The manuscript addresses an important problem of detection of Shiga-like toxins produced by some E. coli strains called by the acronym STEC. The manuscript describes interesting study for testing the CANARY biosensor B-cells, produced as a tool of Stx biodetection.

Major Remarks

Lines 92-94 and 302-305

The information about the elaborated biosensor cells constructed of five different a-STX sequences is very vague and short. In my opinion the precise description of the biosensor cells production should be presented, as it is the basis for this work.

There is no information what was the antigen for the production of a-Stx antibodies: was it toxin or toxoid, what subtype of toxin, produced in which strain of bacteria? Was the whole sequence of heavy and light chains synthesized or only variable domains or their epitope-binding parts?

Was the procedure performed as a commercial service or experimental procedure? It should be described in detail, other procedures in Materials and Methods part are presented with extreme precision. If it is proprietary information it should be stated, but as this is the most important part of the biosensor, it should be described in detail along with the whole procedure and peptide-coding sequences.
The sequences of the antigen-recognizing regions of antibodies should be provided in supplementary data.

The biosensor has not been tested using bacteria not producing Stx, and other than E.coli. I has not been tested on more complex matrices as poisoned food or environmental samples, what limits the knowledge about the specificity of the produced sensor.

Minor remarks
Institution acronyms should be explained or institution protocols (FSIS, FDA BAM)
Shiga toxin is produced by Shigella bacteria. E. coli toxin is called Shiga-like toxin, especially Stx2 shares only partial homology with Shiga toxin. It should be definitely changed in the title to the term “shiga-like toxin” and the difference explained at least at the beginning of the text.

Line 40: STEC are considered BSL-2 not BSL-3 bacteria, complication can occur due to Stx presence, but the infection itself is curable, therefore the statement is false.
Line 41: PCR cannot be used to detect toxin itself, it can be used to detect Stx gene in bacteria
Line 47:290 "ng/kg" kg of what?
line 52: "STEC concentration" Or,, STEC titer" instead of "STEC"

Line 346: bacterial strain number should be provided

line 302: the DNA sequence coding the heavy and light chains...

Author Response

Reviewer 3 comments:

  1. Lines 92-94 and 302-305.

The information about the elaborated biosensor cells constructed of five different a-STX sequences is very vague and short. In my opinion the precise description of the biosensor cells production should be presented, as it is the basis for this work.

There is no information what was the antigen for the production of a-Stx antibodies: was it toxin or toxoid, what subtype of toxin, produced in which strain of bacteria? Was the whole sequence of heavy and light chains synthesized or only variable domains or their epitope-binding parts?

Was the procedure performed as a commercial service or experimental procedure? It should be described in detail, other procedures in Materials and Methods part are presented with extreme precision. If it is proprietary information it should be stated, but as this is the most important part of the biosensor, it should be described in detail along with the whole procedure and peptide-coding sequences.
The sequences of the antigen-recognizing regions of antibodies should be provided in supplementary data.

---  Thank you very much for your comments and suggestions.  The antigen used to generate the Stx mAbs was the Stx2a toxoid (recombinant Stx2a E167R expressed in E.coli) and described in reference 45 [ He et al 2013 Journal of Immunological Methods]. We have more information regarding Stx2a and the generation of the 5 mAbs in the material and methods section 5.2. This manuscript was a collaborative Cooperative Research and Development Agreement (CRADA) between Smiths Detection and the USDA-ARS. Thus the inherent details regarding the biosensor development and biosensors themselves are proprietary information.

  1. The biosensor has not been tested using bacteria not producing Stx, and other than E.coli. I has not been tested on more complex matrices as poisoned food or environmental samples, what limits the knowledge about the specificity of the produced sensor.

 Thank you for the insightful suggestion regarding the bacteria not producing Stx, environmental, or food samples to be evaluated in this assay.  We have evaluated strains without Stx as negative controls in the manuscript in particular ATCC25922 (O6, stx-) and RM7103 (O45, stx-) in Figure 6, Table 2, lines 306-307, and lines 461-463 in section 5.7 of M&M. The complex matrices or environmental sample experiments are not in the current scope of this manuscript which describes the initial development of the STEC assay.  We have another manuscript that relates to the applicability of this STEC biosensor assay for detection of Stx/STEC in contaminated food samples.

Minor remarks
3. Institution acronyms should be explained or institution protocols (FSIS, FDA BAM)

--- Thank you for this comment. We have modified in the abstract at the first mention of the acronyms what they refer to.

  1. Shiga toxin is produced by Shigella bacteria. E. coli toxin is called Shiga-like toxin, especially Stx2 shares only partial homology with Shiga toxin. It should be definitely changed in the title to the term “shiga-like toxin” and the difference explained at least at the beginning of the text.

--- Thank you for the suggestion. We have modified the title to be “ Development of a Rapid and Sensitive CANARY Biosensor Assay for the Detection of Shiga Toxin 2 from Escherichia coli” to give more clarity. We have also briefly described the difference between Shiga toxin and Shiga-like toxin at the beginning of the Introduction and modified to include ...” the Shiga-like toxins (Stxs) are the predominant…” Additionally, we have also modified “Stxs from Escherichia coli are ribosomal inactivating…”

  1. Line 40: STEC are considered BSL-2 not BSL-3 bacteria, complication can occur due to Stx presence, but the infection itself is curable, therefore the statement is false.

--- Thank you for the comment. We have modified the text to give clarity to our original statement “… since there are no FDA-approved therapeutics to treat the complications that can arise from STEC infections ”( line 52-53).

  1. Line 41: PCR cannot be used to detect toxin itself, it can be used to detect Stx gene in bacteria

--- Agreed, we have modified this statement in line 54-55 to state detect Shiga toxins and/or the presence of Stx genes…

  1. Line 47:290 "ng/kg" kg of what?

---  We have added 290 ng of Stx2 per kg of mouse body weight to this line to give clarity (line 60-61).

  1. line 52: "STEC concentration" Or,, STEC titer" instead of "STEC"

--- Agreed, we have modified the new line 66 accordingly with “the STEC titer.”

  1. Line 346: bacterial strain number should be provided

--- Agreed, we have added the bacterial strain number NR-17630 from BEI (line 451) in the manuscript.

  1. line 302: the DNA sequence coding the heavy and light chains...

--- Agreed, we have modified this line (line 399) in the manuscript as suggested.

Round 2

Reviewer 2 Report

Comments and Suggestions for Authors

The manuscript has been revised and my comments have been adequately answered. Except for question #4. I understand that controls were carried out; however, if they are not depicted, at least they should be mentioned in the legend.

Reviewer 3 Report

Comments and Suggestions for Authors

I have no further comments to the manuscript.